# Towards Advancing Real-Time Railroad Inspection Using a Directional Eddy Current Probe

**DOI:** 10.3390/s24206702

**Published:** 2024-10-18

**Authors:** Meirbek Mussatayev, Ruby Kempka, Mohammed Alanesi

**Affiliations:** 1School of Electrical, Electronic and Mechanical Engineering, University of Bristol, Bristol BS8 1TR, UK; 2Kazakhstan Institute of Non-Destructive Evaluation LLP, 59, Tole bi Street (KBTU), Almaty 050005/A05H1T2, Kazakhstan; 3Department of Mechanical Engineering, The University of Sheffield, Mappin Street, Sheffield S1 3JD, UK; r.kempka@sheffield.ac.uk; 4Department of Intelligent Manufacturing Engineering, Guilin University of Electronic Technology, Guilin 541004, China

**Keywords:** Eddy Current Testing, lift-off, real-time inspection, probe optimisation, signal-to-noise ratio, non-destructive testing

## Abstract

In the field of railroad safety, the effective detection of surface cracks is critical, necessitating reliable, high-speed, non-destructive testing (NDT) methods. This study introduces a hybrid Eddy Current Testing (ECT) probe, specifically engineered for railroad inspection, to address the common issue of “lift-off noise” due to varying distances between the probe and the test material. Unlike traditional ECT methods, this probe integrates transmit and differential receiver (Tx-dRx) coils, aiming to enhance detection sensitivity and minimise the lift-off impact. The study optimises ECT probes employing different transmitter coils, emphasising three main objectives: (a) quantitatively evaluating each probe using signal-to-noise ratio (SNR) and outlining a real-time data-processing algorithm based on SNR methodology; (b) exploring the frequency range proximal to the electrical resonance of the receiver coil; and (c) examining sensitivity variations across varying lift-off distances. The experimental outcomes indicate that the newly designed probe with a figure-8 shaped transmitter coil significantly improves sensitivity in detecting surface cracks on railroads. It achieves an impressive SNR exceeding 100 for defects with minimal dimensions of 1 mm in width and depth. The simulation results closely align with experimental findings, validating the investigation of the optimal operational frequency and lift-off distance for selected probe performance, which are determined to be 0.3 MHz and 1 mm, respectively. The realisation of this project would lead to notable advancements in enhancing railroad safety by improving the efficiency of crack detection.

## 1. Introduction

Non-destructive testing (NDT) plays a crucial role in ensuring the safety and reliability of engineering structures across various sectors, including automotive, aerospace, and civil engineering. A critical component in the railway system is the railway track, which is susceptible to various types of cracks. Current standards for railway inspection predominantly rely on visual testing (VT) methods, which are time-consuming and qualitative, and VT remains limited by top surface inspection and the need for expertise. Monitoring surface and sub-surface defects, including burns, squat-like rail cracks, and other types of local damages, costs millions of pounds every year in the U.K. [1,2]. Railway inspection and the capability for the real-time inspection of these defects present significant challenges, with only a limited array of NDT techniques available. Among these, Ultrasonic Testing (UT) is frequently utilised due to its ability to penetrate the bulk material of the track, particularly in the rail head and web. However, this technique encounters a notable limitation in identifying both surface and subsurface anomalies smaller than 4 mm in depth during high-speed inspections [3,4]. While it is possible to detect surface imperfections through the reflective echoes of ultrasonic waves once they reach a particular size, UT is not capable of assessing the extent of these defects, nor can it differentiate among various defect categories [5]. Thus, there is an urgent need for a more efficient and reliable surface and sub-surface inspection method to ensure the structural integrity of rail tracks. Consequently, ECT has been identified as an appropriate NDT technique for in-line railway inspection. To realise this application and facilitate the full automation of the inspection process, the integration of various distinct technologies is imperative. This integration is essential to address the complex requirements of real-time, efficient, and accurate railway inspection. This project aims to significantly enhance efficiency by implementing real-time on-train inspection technology, as illustrated in Figure 1. The primary challenge in measurements of this nature is not inherently linked to the limitations of sensor technology but rather pertains to the management and processing of the data collected [6].

Many attempts to embed the similar EC systems for online detection and location of rails defects have been published before [7,8,9]. However, the lack of available EC probe optimisation studies for real-time inspection [10], coupled with the need for improved data management options, motivates the present investigation. A specific online data-processing algorithm is provided to effectively manage the huge number of data associated with the nature of this type of inspection. During each inspection, the system aboard the train accumulates a substantial volume of data, within which only a fraction are of significant relevance. Identifying and extracting these pertinent data segments is crucial for determining their precise locations along the track. Typically, specialised research groups are dedicated to the management of such data, employing strategies that involve either transmitting the data for remote analysis or conducting local analysis onboard to detect anomalies that surpass predefined thresholds. Despite the fact that, based on the proposed improved system shown in Figure 1, this project needs to characterize the different types of actual types of railroads damage, this initial research is limited to the search and identification parts.

This initial research aims to optimise the proposed ECT probe designs to develop an inspection system integrated within on-train inspection technology capable of real-time inspection in dynamic conditions. Further studies will be more focused on investigating characterisation of localised rail track defects, particularly squats, smuds, and rolling contact fatigue (RCF) cracks. The goal is to optimise the sensor design, enhancing its sensitivity to these specific defects under varying inspection conditions, including factors like speed and vibration. Another challenge is posed by strongly ferromagnetic steels in ECT, particularly in the context of rail steels with high coercivity. Compelling instances of overcoming this challenge are demonstrated through the development of ECT probes for numerous rail-inspection systems [11,12,13], encompassing both ECT and UT [14,15,16], as well as integrated ECT on-train inspection technologies [17,18,19,20]. High coercivity can be addressed through appropriate EC probe design and optimisation. For instance, Anandika et al. [21] investigated the sensitivity of the conventional differential eddy current (EC) mode for detecting defects in rail samples. The study provided information on the coil distance width and operational frequency of the probe; however, details such as the number of turns, the dimensions, and the inductance of the coil were not included. The choice of a 5.5 mm coil distance and a 70 kHz operational frequency was not supported by references to previous studies, which leaves the optimisation of the test probe uncertain. Despite the advantages of differential-mode EC testing, including insensitivity to lift-off and temperature variations, as well as the ability to detect large-scale structural variations, the lack of comprehensive probe details may impact the reproducibility and reliability of the results. Additionally, the authors noted that the manual movement of the probe during some tests introduced variability into the EC scan’s results. However, a similar probe developed by the authors of [13] demonstrated that after optimising the operational frequency and eliminating additional sources of noise (such as manual inspection), the AC bridge measurements successfully detected rail defects. Thus, the optimisation of the EC probe is crucial for resolving issues related to penetration depth due to the high coercivity of rail steel. In addition, to address the issue of defect detection, the finite element method was applied to predict the detectability of targeted defects using high-coercivity soft iron material as the core. In [22], the authors tested conventional transmit–receive EC sensors for simulated rail defect detection, highlighting challenges in extracting defect-modulated signals. They found that the detection coil’s output is predominantly influenced by direct induced voltage from the excitation coil, making it difficult to discern voltage variations caused by defect-induced EC changes buried within the large baseline signal. Additionally, variations in the distance between the sensor and the rail due to vibrations and bumping introduce a “lift-off” effect, further complicating defect characterisation. The lift-off distance for online rail defect inspection using ECT typically ranges between 1 and 3 mm [7,23]. This range is optimal to balance sensitivity and minimise noise from surface roughness and other extraneous factors. However, dynamic measurements with the selected probe are planned to be conducted using a trolley. Thus, significant deviations in surface roughness and other extraneous factors can be controlled to some extent before scanning the section of the rail. Another study [24] highlights the significant advancements in lift-off suppression techniques for ECT, emphasising their relevance and applicability to current rail inspection specifications by enhancing sensitivity and accuracy under challenging lift-off conditions. Proper probe optimisation to enhance sensitivity to the feature of interest is advisable, aiming to maintain relatively higher sensitivity for real-life scenario experiments where variation in lift-off is a dominant factor affecting probe sensitivity.

In response to these challenges, the study has developed new transmit and differential receiver (Tx-dRx) coils, tailored to fulfil the unique requirements of real-time, accurate rail track inspection. A substantial body of research has been dedicated to the optimisation of EC probes, with various studies focusing on enhancing their efficacy in diverse applications.

In [25], the authors detail the optimisation of a low-frequency EC technique for internal defect detection in steel structures, analysing a custom-designed magnetic sensor system in tandem with finite element modeling (FEM) results. This study explores the fine-tuning of probes using magnetoresistive sensors, supported by simulations. Reference [26] delineates the development of a flexible planar EC sensor array, targeting microcrack inspection in critical airplane components. The study details the sensor design and measurement mechanics and correlates these with FEM outcomes. In a significant advancement, the authors of [27] present a differential coupling double-layer coil for ECT, achieving notable improvements in sensitivity and lift-off tolerance. This coil’s innovative double-layer structure and differential coupling energy mechanism demonstrate the potential of strategic coil design and frequency optimisation in overcoming traditional limitations of EC probe sensitivity, especially in scenarios with high lift-off. An extended literature review of applications and advantages of planar rectangular receiver in ECT provided in this study. Further progress is reported in [28], which proposes a novel figure-8-shaped coil for transmitter–receiver (T-R) probes. This unique design effectively counters signal distortion due to lift-off variation and maintains consistent output when aligned with the CFRP’s fibre orientation. The probe is characterised by its insensitivity to lift-off variations and enhanced sensitivity to defects in CFRPs. Lastly, the authors of [29] present an analytical model for a figure-8-shaped coil comprising two oblique elliptical coils. This model enables the manipulation of the electromagnetic concentrative region and the EC density. Adjustments in the elliptical shape or the spread angle between the coils lead to intensified and expanded eddy currents (ECs), concentrating under the coil’s symmetric centre. This innovative design and analytical methodology significantly elevate the accuracy in detecting conductive material defects, marking a pivotal development in NDT methods. While each study substantially enriches the field of ECT probe optimisation and NDT, their practical application varies based on specific conditions, material types, and defect characteristics. These advancements lay the groundwork for exploring the applicability of novel EC probes in railroad inspection.

The overarching aim of this project is to develop EC sensors tailored for the real-time monitoring of railway track surfaces and to create an online system for detecting and automatically identifying rail defects with directional EC probes. The authors of [9] accomplished this by evaluating the Wavelet Power Spectrum using a convolutional neural network. The improved optimisation of EC sensors explored in this initial research will help to identify some crucial parameters such as optimum frequency and lift-off distance versus the selected EC probe sensitivity. Consequently, the optimum parameters for a dynamic measurement setup will be identified through static measurements performed in this study. The objective of this paper is the optimisation of ECT sensors tailored for the real-time surface inspection of rail tracks, exploring design parameters of a novel directional ECT probe for detecting surface-breaking cracks and common railway flaws.

## 2. Methodology

The development of the proposed inspection system, as depicted in Figure 1, necessitates comprehensive studies to verify the feasibility and reliability of integrating the probe into the real-time, on-train inspection technology. This integration must account for variations in speed and vibrations from both the train and the rail tracks, necessitating probe optimisation. However, the current experimental setup is primarily aimed at optimising the probe design for heightened sensitivity to common defects in ferromagnetic materials, rather than replicating the exact operational environment. The directional probe design previously outlined in references [29] has not yet been tested on metallic components. Studies in [30,31,32] have demonstrated optimum sensitivity beyond the receiver’s resonance for CFRP materials, which now requires validation for ferromagnetic materials. Therefore, our methodology involves multiple studies focusing on real-time EC probe optimisation. This research utilises the SNR for quantitative evaluation, comparing the sensitivity of the best probe configuration from earlier research [31] with the two novel sensor designs proposed in this study. Investigating the optimal frequency is crucial for boosting sensitivity to specific defects. This study delves into the proximity frequency range, resonating with the receiver coil to elevate SNR levels. Similar research [6] analysed phase shifts with frequency variations to identify the optimal operational frequency, noting that increased phase shifts improve defect depth resolution and decrease detection errors. Another critical aspect of this research is determining the optimal lift-off distance, ranging from 0.25 to 1 mm, as lift-off variation is a key parameter before in situ inspection.

This involves balancing sensitivity against lift-off distance, an essential factor in EC probe optimisation. Related research [7] conducted fixed-distance (1 mm) inspections from the rail track surface by integrating EC instrumentation into a grinding train for the early detection of surface damages. This highlights the importance of understanding the trade-offs between sensitivity and lift-off distance for effective EC probe deployment in real-world applications.

### 2.1. Sensor Design

The paper presents a frequency-selection study focusing on the three sensor designs depicted in Figure 2. To minimise the liftoff noise, two differentially winded adjacent rectangular sensors were designed and manufactured, and to increase the induced ECs density and concentrative area within the material single transmitter, special figure-8-shaped elliptical and four figure-8-shaped (+point) transmitter coils were designed and manufactured. The initial design utilises singular excitation transmitter coils, while subsequent modifications involve altering the aspect ratio of the previous non-uniform configuration [31], as depicted in Figure 2a, tailored for the inspection requirements of railroad infrastructure. Drawing upon principles from EC theory, the inclusion of winding +point [6,33] and figure-8-shaped [28] configurations aims to amplify the sensitivity of pickup signals. However, it is noteworthy that similar transmitter designs have not been previously manufactured to enhance the sensitivity of the EC probe for the specific needs of the targeted application.

### 2.2. Transmitter Coil Design

To optimise the performance under the distributed magnetic field of the transmitter coil, it is recommended that the width of the coil be equal to or exceed the length of the receiver coil. Increasing the number of turns in the transmitter coil strengthens the magnetic field but raises the inductance, consequently lowering the resonance frequency. The fabrication of transmitter coils with 62, 72, and 66 turns on ferrite, corresponding to single, +point, and figure-8-shaped configurations, respectively, ensures the generation of a suitably robust magnetic field to produce the desired excitation frequency signal. Table 1 provides transmitter and receiver coils’ parameters. The peak frequencies of the transmitter coil vary based on winding quality and turn count. Figure 3a–c show +point, figure-8-shaped, and single winding methods, respectively, along with their dimensions and resonance frequencies. The transmitter coils’ impedance spectra were assessed employing a Network Analyzer (TE3001, TrewMac Systems Ptd Ltd, Morphettville, Australia). Resonant frequencies of approximately 1.1, 0.6, and 1 MHz were observed for the single, +point, and figure-8-shaped transmitters, respectively.

### 2.3. Planar Rectangular Receiver Coils

The planar rectangular pickup coils’ advantages and some of their limitations, along with their application challenges, are reviewed by the study in [27]. The consistency of detected signals and the flexibility to manufacture with various dimensions to accommodate different track widths make printed PCB-based technology an ideal candidate for manufacturing. The original resonance of the receiver coil is 10.3 MHz, but the external capacitors (14.7 nF) were applied to tune the resonant frequency of the sensor coils to around 0.9 MHz. Thus, the sensitivity was enhanced for a targeted defect. The sensor parameters and dimensions are presented in Table 2 and Figure 3d, respectively.

### 2.4. Amplifier

In ECT NDE, limitations arise from noise sources like sample electromagnetic property variations, vibration, temperature changes, and probe lift-off and tilt. Utilising SNR as a metric for probe performance evaluation is advisable. Figure 4 illustrates the fundamental components of the amplifier and the coil input circuits. The low-pass filtering effect of the feedback amplifiers is characterised by the closed-loop bandwidth, which is the unity gain-bandwidth product divided by the closed-loop gain. While further reducing bandwidth at a specific gain value is feasible, it introduces phase lags at target frequencies, impacting gain. In the simplified circuit diagram, all external circuitry is situated left of the “Input 1” and “Input 2” terminals, with onboard components to their right. Coils L1 and L2 function as receiver coils, inducing emfs in series opposition, yielding a differential voltage output. L3 and L4, the drive coils, wound in a figure-8 pattern around ferrite bar cores, generate opposing AC fields at adjacent poles. The drive coil tuning capacitor, chosen for resonance, amplifies coil currents and voltages, enhancing sensitivity and SNR. The combined output of the receiver coils serves as a differential input signal for the amplifier, connected to the Input 1 and 2 terminals. The input circuit features switch-selectable capacitors for coil tuning and various loading resistors, with the option to disconnect the centre ground connection. Implementing a non-earth-referenced input via a signal isolation transformer is preferred, offering cost-effective common mode rejection. The amplifier circuit, inclusive of the isolation transformer, delivers a gain of 10 (or +20 dB) into an infinite resistance load, with a flat frequency response between 100 kHz and 2 MHz. Beyond 5 MHz, the gain reduces to 19.3 dB with a 40° phase lag, with a 3 dB drop at 11.25 MHz. For remote signal transmission, a 50 ohm coaxial cable is advised, with a 50 ohm termination for cables longer than approximately 3 m, typically reducing gain by 6 dB. Capacitor switchers were integrated into the amplifier board to adjust the receiver coil’s resonant frequency within a specified range. Equation 1 provides a formula for calculating the required capacitance for the desired resonance based on probe coil inductance and target frequency.
(1)$$C=\frac{1}{w_{0}^{2}L}$$
where $\omega_{0}=2\pi f_{0}$. Here, $f_{0}$ represents the desired resonant frequency (1.5 MHz), C is the tuning capacitance in farads, and L denotes the probe inductance (1.0 × 10^−6^ H). The signal-to-noise ratio (SNR) quantifies the proportion of the desired signal to the level of background noise. It is a pivotal metric in the evaluation of system performance, particularly in the context of signal processing.

### 2.5. Working Principle and Signal Processing of Sensors

The measurement mechanism proposed in this study is based on the principle of electromagnetic induction. An alternating current supplied to the coil generates a changing magnetic field around the wires, inducing eddy currents in nearby conductive materials. These induced currents create their secondary magnetic fields, flowing opposite to the primary magnetic field, and contain information about the conductivity and permeability of the tested material. By monitoring changes in receiver impedance, the presence of defects can be detected. The measurement process occurs in real time and is monitored online via the Matlab R2024a environment, with further post-processing conducted. The real-time inspection demonstration of EC probes equipped with both point and figure-8-shaped transmitters can be found in the Appendix A. This demonstration video illustrates the experimental scanning process and the detection of anomalous signals, indicating the presence of rail defects. Additional details on post-processing methods can be found in Appendix A1 of ref. [31].

### 2.6. Estimation Method of SNR

The evaluation of the three probe design sensitivities and identification in Figure 1 was quantitatively assessed using SNR. Figure 5 depicts the proposed methodology for identification based on a 0.4 MHz exemplar sensor signal using an EC probe with a figure-8-shaped transmitter coil.

Data collection began from 0 to 50 mm over a large, non-damaged area of the rail track (as shown in Figure 5). These data are utilised for automatic calculation of twice the root-mean-squared (RMS) value of the background structural noise as a threshold to distinguish sensor signals (indicated by the horizontal dashed line in Figure 5). Any signal exceeding twice the mean RMS noise voltage is considered a sensor signal and undergoes peak value detection. Upon passing the defective zone, the algorithm automatically provides information about the estimated SNR calculated using Equation (2):(2)$$C=\frac{V_{max}+abs(V_{min})}{2N_{rms}}$$
where V_max_ represents the maximum observed signal voltage, V_min_ denotes the minimum observed signal voltage, and N_rms_ is the root mean square value of the noise level. Here, the signal-to-noise ratio (SNR) quantifies the proportion of the desired signal to the level of background noise and is a pivotal metric in the evaluation of system performance, particularly in the context of signal processing.

### 2.7. Finite Element Modelling of Three Probe Configurations

A comprehensive FEM investigation was executed using COMSOL 6.1. The simulation speed was carefully calibrated to balance computational efficiency and accuracy, which is critical for practical applications. The study leveraged insights from Chen et al. [26], utilising a differential coupling double-layer coil approach to enhance testing capabilities, especially in high-lift-off scenarios.

### 2.8. Modeling Method

The virtual scanning model was executed employing COMSOL 6.1, as meticulously illustrated in Figure 6. The model spans the *x*-axis from −20 mm to 20 mm, encompassing both the defective area (ranging from −0.5 mm to 0.5 mm) and the surrounding non-defective zones, which exhibit structural variations. The detailed FEM setup parameters are presented in Table 3.

#### 2.8.1. Mesh Configuration and Finite Elements

To ensure the accuracy of the simulation results, different finite elements were utilized for various regions within the FEM model. Fine mesh elements were used near the defect zones to capture detailed EC interactions, while coarser elements were applied to the non-defective surrounding areas to optimise the computational efficiency. The number of elements varied depending on the region, with a higher concentration of elements in the critical defect areas to enhance detection precision. The mesh size had a significant impact on the simulation results. A finer mesh size increased the resolution and accuracy of the defect detection but also required more computational resources. Conversely, a coarser mesh reduced the computational load but could potentially overlook smaller defect features. Therefore, a balance was achieved by adjusting the mesh density according to the specific requirements of different regions within the model.

#### 2.8.2. Defect Type and Meshing Criteria

The criterion for meshing involved ensuring that the mesh elements were sufficiently small to accurately represent the defect features. In this study, the defect type specified was a crack. This allowed for the precise simulation of the EC distributions and their interactions with the crack defect. The accurate representation of cracks was essential for evaluating the sensitivity and performance of the different probe configurations. This method proved effective in increasing the sensitivity and accuracy in defect detection, which is critical for practical NDT applications.

### 2.9. Probe Configuration Analysis

The primary objective of the probe configuration analysis is to rigorously assess the efficacy of various probe configurations—namely “Singular”, “+Point”, and “Figure-8”—in detecting defects through induced ECs. This comparative evaluation seeks to ascertain which configuration demonstrates superior sensitivity and precision in defect detection, thereby informing the optimal design for practical applications in NDT.

Figure 7i provides a comprehensive overview of the FEM mesh, delineating the air domain and transmitter coils within the “Singular”, “+Point”, and “Figure-8” configurations. Additionally, it includes a detailed close-up of the transmitter coils positioned adjacent to the rail defect. The resultant EC distributions generated by the “Singular”, “+Point”, and “Figure-8” configurations are depicted in Figure 7ii(a–c), respectively.

It is noteworthy that all configurations induce non-uniform current distributions. Among these, the EC probe with a figure-8-shaped transmitter coil exhibits the highest EC density. This heightened density can be attributed to the increased elliptical factors of the figure-8 transmitter coil, which facilitates the induction of substantial ECs densities within defect regions, as illustrated in Figure 7ii(c). This characteristic effectively mitigates structural influences, enhancing defect detectability. In contrast, the EC densities observed in Figure 7ii(b) from the +Point transmitter configuration indicate that this probe might achieve better performance if oriented at a −45-degree angle, thereby potentially improving its detection capabilities.

## 3. Experiments

### 3.1. Rail Track with an Induced Artificial Defect

A series of sore cuts introduced on the top surface of the rail track specimen for the proposed inspection system’s optimisation study as is shown in Figure 8a. The material of the sample is made from the standard R260 grade rails. The probe capabilities can be evaluated best by inspecting one artificial defect over the relatively large non-defected area. The dimensions of the sore cut are 40 × 1.0 × 0.8 mm.

### 3.2. Measurement Set-Up

The experimental set-up is illustrated in Figure 8b. The main magnetic field generated by the transmitter coil induces ECs within the rail track sample, resulting in the generation of a secondary magnetic field. The secondary magnetic fields, which vary over time, cause a voltage to be induced in the receiver coils. The presence of a near-surface defect on the rail disrupts the local distribution of the ECs, causing a slight alteration in the induced voltage detected by the coil. The set-up comprises a handyscope HS5 (TiePie Engineering, Sneek, Netherlands) generating a sinusoidal signal, linked to a Holland current source (Sonemat Ltd., Coventry, UK) and the DC power supply HY 3003 (Mastech, Hong Kong, China). One output from the current source supplies a consistent current to the transmitter coil for magnetic field excitation. The other output is connected to the handyscope to serve as a stable voltage reference for voltage difference comparison with received pick-up signals from the probe. The power supply provides a ±9.5 V DC supply to the differential amplifier. Real-time control and post-processing of measurements are handled by a MATLAB script interfacing the handyscope with the computer.

## 4. Results

### 4.1. Frequency Selection Study

The experimental setup involved conducting at least three repeated independent measurements across a 200 mm range for each specimen to enhance precision in real time using MATLAB. The movement increment was set to 0.125 mm, resulting in a maximum of 1600 data points. The collected data underwent post-processing to detrend signal amplitudes, followed by smoothing using MATLAB’s built-in filtering and the subsequent calculation of the SNR.

A symmetrical sensor signal from the differential receiver coils, obtained as they scan over a surface defect, is illustrated in Figure 9. The structural variation of rail steel before the 40 mm mark on the EC raw plot represents the differential voltage response measured by the probe as it scans over an unflawed region of the material. As the probe moves toward the defect zone (toward the right in the figure, up to the 55 mm position), only the first of the two adjacent coils senses the defect, causing a change in the probe’s total impedance. When the probe moves further right, the nearly non-defective voltage value is recovered when the defect is positioned centrally between the two coils, as both coils are equally but oppositely affected by the presence of the crack. As the probe continues to move right, only the following coil of the two adjacent receiver coils senses the defect. Since the coils are wound in opposite directions, the sensor signal in Figure 9 has a symmetric shape.

The experimental investigation into optimum frequency selection for EC probes utilised frequency ranges from 0.2 to 0.7 MHz with a single transmitter, from 0.4 to 1.3 MHz with a +point transmitter, and from 0.15 to 0.6 MHz with a figure-8 shaped probe with 0.35 V to drive the transmitter coil. The selection of frequency ranges was influenced by observed noise levels. Higher frequency ranges near the resonant frequency of the receiver coil were avoided due to challenges posed by factors such as magnetic hysteresis, complex microstructure, and high permeability in ferromagnetic materials, except for the EC probe with the +point transmitter. The EC method often fails because it cannot distinguish between the impacts of conductivity and permeability on probe impedance, except at frequencies generally below the typical operating range of most probes. By increasing the applied voltage from the handyscope from 0.35 V_pp_ to higher values, the relatively high inductance of the transmitter coil (296 microhenries) causes the measurement system to reach the point where the monitoring voltage across the transmitter coil starts to clip earlier. This clipping occurs because the voltage generated across the transmitter coils, with a fixed current passing through them, reaches the limit set by a Holland current source at a certain frequency. Optimising EC probes necessitates careful control of factors affecting measurement noise levels and an adjustment of the resonant frequency. Capacitors of 14.7 nF were used to tune the receiver coil’s resonant frequency, with each coil having an inductance of approximately 2.35 µH, tuned for resonance at the operational frequency. Specific resonant frequency could be achieved by calculating the required capacitance using Thompson’s formula. The sensor signals obtained from FEM simulations exhibit variations compared to the experimental data. The discrepancies were observed in the voltage values of the figure-8 shaped transmitter coil, where the experimental values exceeded the simulated voltage values. This difference can be attributed to the high-permeability material (soft iron) assumed in the modeling, while the experimental conditions may differ due to conductivity variations in rail steel. Additionally, variations in the morphology of sensor signals from the experimental data using the +point transmitter compared to the simulation could be due to differences in scanning speed [27]. Given that the speed is beyond the scope of this study, it will be considered in subsequent investigations for result comparison. In each row, Figure 9i displays the raw EC data for each probe configuration, and (ii) shows simulated corresponding differential voltage outputs. Overall agreement was observed between the induced voltage values for the FEM results and the experiments where the selected figure-8 shaped probe outperforms the results of other two probes. The SNR assessment results (see Figure 10) indicate that Probe 1 generates a stronger ECs flow across the scanning direction, leading to consistently elevated sensor signals at frequencies around 0.3 MHz. The findings from both the simulation and experimental studies highlights the importance of ECT probe optimisation before employing any machine learning approach. However, discrepancies are observed, particularly in the clear differences in the sensor signal shape and the absolute voltage value for the figure-8 shaped probe. These discrepancies arise because the simulation study relied on a higher coercivity material, as referenced in Table 3. The accurate experimental parameters have not been precisely retrieved, which is currently under investigation. Figure 10 illustrates the SNR estimation outcomes around the sensor coil’s resonant frequency, demonstrating improved signal strength and reduced uncertainty across various frequency ranges. All scan positions except those marked by the vertical dashed lines are used as undamaged sections of the structural background rail to assess the RMS of noise for each measurement. To evaluate the probe performance, the mean RMS noise value is multiplied by two to establish the noise threshold level (indicated by the horizontal dashed line in Figure 9i(a–c)). Among the probes, the one with the figure-8 shaped exhibited the lowest structural noise value, while the probe with the singular excitation showed the highest. Among the three EC probes, the figure-8 shaped configuration outperformed the other two, with nearly triple the SNR, with the maximum SNR recorded at 105 at 0.25 MHz. In this probe configuration, the elongated figure-8 shaped coils drive current across the scanning direction, as depicted in Figure 7, and we observe a clear increase in current density at the defective zone. This phenomenon arises from the generation of currents in both the clockwise and counterclockwise directions. The singular EC probe exhibited real-time inspection capability, achieving an SNR of approximately 13 at 0.5 MHz, while the +point transmitter probe doubled the performance, achieving around 27 SNR at 0.6 MHz. Further investigation into varying lift-off distances and their impact on ECT sensitivity will focus on the figure-8 shaped probe.

### 4.2. Lift-Off Distance and Optimum Probe Sensitivity

Understanding a selected probe’s sensitivity to targeted defects, particularly concerning variations in lift-off distance, is crucial before implementing the sensor in real-world settings. Previous studies consistently maintained a 0.25 mm distance across all measurements. However, in [7] the research applied sensor integration in a similar environment with a 1 mm lift-off distance. Therefore, to explore probe sensitivity across a range of 0.25 to 1 mm, a minimum of three independent measurements were conducted, varying both frequency and lift-off distance. The rationale for maintaining a constant frequency range was to evaluate the noise level at different lift-off distances.

This approach is critical for future studies, where the integration of this probe configuration might be necessary under varying operational frequencies. MATLAB R2024a software was utilised to ensure real-time accuracy in data acquisition and analysis. Figure 11a–c present the selected probe sensitivity profile, demonstrating its differential voltage response as a function of varying lift-off distances. A notable trend observed was a significant decrease in the recorded differential voltage as the distance from the rail surface increased within the range of 0.5 to 1 mm. In addition to the experimental data, FEM simulations were conducted to assess the impact of lift-off distance on probe performance. The simulation results, presented in Figure 11d–f, demonstrate that the selected probe configuration exhibits a similar trend to the experimental results as the lift-off distance increases. A significant drop in differential voltage was observed at the maximum lift-off distance of 1 mm. To facilitate a quantitative comparison of the experimental results and assess the individual contributions of influential factors such as signal and coherent structural noise, the raw EC data presented in Figure 11 underwent further post-processing, following the methodology described by [31].

To evaluate the selected probe’s performance at different frequencies, the RMS of structural noise was first analysed across varying lift-off distances to subsequently calculate the SNR, as shown in Figure 12a. At a lift-off distance of 0.5 mm, the SNR reached its peak at approximately 105. However, it is worth noting that the uncertainty represented by the error bars was relatively higher than those observed at the 0.25 mm lift-off distance, as depicted in Figure 12b. At 0.3 MHz, the SNR exhibited a gradual increase across all three distances, indicating enhanced SNR. This trend coincided with relatively lower levels of structural noise observed at this optimal frequency. At the 1 mm lift-off distance, the lowest SNR values with higher uncertainty were recorded after the 0.4 MHz frequency. The observed phenomenon was attributed to the higher levels of coherent noise observed between 0.4 MHz and 0.6 MHz, as is demonstrated in Figure 12a. Commencing at the upper frequency ranges of 0.35–0.6 MHz, a notable decline in SNR of the selected probe was observed for all lift-off distance ranges, as depicted in Figure 12b. Even when the lift-off distance was increased to 1 mm, the probe exhibited the lowest sensitivity in terms of voltage magnitude (see Figure 11c), yet it maintained a relatively high SNR between 40 and 77. The mean SNR, signal voltage values from three lift-off distance measurements were analysed, along with the RMS noise associated with each, and plotted as a function of lift-off distance at a frequency of 0.3 MHz, as illustrated in Figure 12c. The obtained results indicate that, in contrast to the peak-to-peak signal voltage, structural noise emerges as the most influential factor contributing to the observed high level of sensitivity. These results indicate that the EC probe featuring a figure-8 shaped transmitter demonstrates a relatively high SNR at a lift-off distance of 1 mm and a frequency of 0.3 MHz. Notably, this configuration maintains excellent detection sensitivity, with an SNR exceeding 75, within a 1 mm lift-off range.

The differential receiver coils used in the proposed measurement mechanism are designed not to minimise the impact of lift-off on defect sensitivity but to exploit the insensitivity of the differential mode to lift-off variations, which can introduce errors in EC measurements. Abdalla et al. [10] pointed out the limitations of ECT related to lift-off, particularly in reflection mode: As the lift-off distance increases, the interaction between the magnetic field and the material weakens, making it more challenging for the receiver coil to detect changes in the impedance caused by the interaction of the primary magnetic field with the ECs in the material. The electromotive force that the receiver coil detects is a result of this interaction. This issue arises because the ECs themselves become less intense at greater distances, reducing the strength of the secondary magnetic field that the receiver coil needs to detect.

The initial EC probe design combined a +Point transmitter coil with two differentially hand-wound meander receiver coils. Initial tests used circular hand-wound differential coils before finalising the design on a PCB board with a 0.5 mm wire and a 6.5 mm coil diameter, as shown in Figure 13a. The measurement setup involved placing the sample on a vibrating linear stage while the EC probe remained static at a 1 mm lift-off. Despite inconsistencies in receiver coil design, preliminary results were promising. Stage vibration, caused by the sample’s weight after each movement increment, allowed only one repeat measurement, as shown in Figure 13b. The reduction in differential voltage was attributed to variations in the depth of artificially induced defects, with the lowest voltage corresponding to a defect depth of around 1 mm. Since the same Tx-dRx measurement mechanism was applied throughout all studies, this insensitivity to lift-off variation test results demonstrates the mechanism’s reliability, robustness, and effectiveness in real-world scenarios before dynamic testing.

### 4.3. Experimental Validation of Probe Sensitivity for Local Rail Damage

To assess the performance of the selected EC probe, the figure-8 shaped probe, for artificial local defects, a steel plate of the same grade was fabricated. The specimen, measuring 155 mm × 80 mm × 21 mm, featured five drilled holes with a diameter of 1 mm, positioned as depicted in Figure 14. Three initial defects, situated from the left edge as illustrated in Figure 14, were examined at operational frequencies of 0.25, 0.3, and 0.35 MHz. Due to the arrangement of receivers in a differential EC mode with a width of 7.5 each, the sensor signals from the initial holes were symmetrical, as shown in Figure 15. However, with a distance of 9 mm between each hole, subsequent scans covered the other two defects simultaneously. Moving further, sensors placed over the first and second holes yielded pickup signals that, when overlapping over the defective zone, formed the first half of an “M”-shaped sensor signal. Subsequently, scanning over the third defect completed the second half of the “M”-shaped sensor signals.

The sensitivity of the selected probe during scanning over each defect, along with its estimated SNR and the contribution of structural noise, were evaluated independently, as depicted in Figure 16. The estimated structural noise was relatively higher due to edge effects, resulting in lower SNR. Despite the noise level, the selected probe exhibited slightly better performance around 0.35 MHz, achieving SNRs of 7.5 and 12.5 for the first and second/third defects, respectively.

### 4.4. Experimental Validation of Probe Sensitivity to Studs

The experimental results from railroad inspection are crucial for validating the effectiveness of the selected EC probe. The results presented in previous sections are based on artificially induced defects in a material with an undamaged, high-quality surface. For this study, a real rail section with local damage, measuring 128 mm in length, was prepared. The material has a carbon content in the range of 0.87–0.97% by mass in its liquid state. In real rail sections with induced local damage, such as studs (see Figure 17a), major cracks develop in the direction of traffic, although the propagation mechanism remains under investigation. Some cracks form at approximately 20 degrees, but there is no consistent pattern, leading to variations in crack depth and angle along with high roughness in the undamaged areas of the specimen. The defects typically have a “U” shape and can penetrate to depths of 3–6 mm in the rail, and the defect formation is extensively studied in [34]. Studs pose a risk because the rail beneath them cannot be effectively tested using ultrasonic NDE from the running surface, except with specialised equipment [34].

Two repeat scans were performed on the defective sample, with the results plotted in Figure 17b. The proposed probe exhibits excellent sensitivity at an optimum frequency to both studs and non-defected areas of the specimen, achieving SNR of 15 and 14.8, indicating high repeatability. The rough surface of the real rail sample is evident in the EC scan results from 0 to 54 mm. Both receivers detected the defected zone, as indicated by the peaks between the vertical dashed lines in Figure 17b. The noise threshold, represented by the vertical dashed line, shows the mean RMS noise level multiplied by two.

The overall agreement was observed between the induced voltage values for the FEM results and the experiments, where the selected EC probe with an 8-shaped transmitter coil outperforms the results of other two probes. The SNR assessment results (see Figure 10) indicate that Probe 1 generates a stronger EC flow across the scanning direction, leading to consistently elevated sensor signals at frequencies around 0.3 MHz. This outcome aligns with analytical calculations for the elongated figure-8 shaped transmitter [29], which demonstrated that increasing the spread angle of two elliptical coils creates two additional reverse concentrative areas on either side of the *x*-axis (see Figure 7ii(c)). This configuration enhances the sensitivity of differential voltage comparison within the same area using rectangular-shaped receivers. The strong magnetic field concentrates between the two elongated coils, covering the entire receiver area with a relatively uniform field, thus enhancing the detectability for receivers and inducing significant disturbances in eddy current flow. Consequently, the precise alignment of the two elongated transmitters above the differential receiver coils is essential for achieving higher probe sensitivity. The findings from both simulation and experimental studies highlights the importance of ECT probe optimisation before employing any machine learning approach.

Comprehensive field trials should be conducted to validate the technology’s performance under real-world conditions. The equipment setup is designed to be portable, enabling easy deployment in the field. This can be achieved by developing a Holland current source board to replace the existing bulkier EC instrumentation used in the experimental setup (see Figure 8b). The new Holland current source board prototype must be thoroughly tested with extended cable lengths for the pickup signal and voltage reference and integrated into an existing probe box. This testing will help identify any issues related to signal integrity or power loss over longer distances. The system should have a reliable power source, such as rechargeable batteries. The selected probe will be mounted on a 3D-printed tooling mechanism, allowing for individual positioning on the measurement tracks attached just behind the wheels of the measuring trolley. To detect local damages on the rail track, the selected probe in the rail test vehicles will be positioned in the middle of the railhead, covering the entire width of the material with one scan. The proposed technology can be made highly usable in the field, providing reliable and accurate measurements for rail track inspection and other applications.

## 5. Conclusions

This study explored various combinations of excitation coil designs in conjunction with a rectangular planar receiver coil, assessing their potential integration into on-train inspection technology for a real-time, high-speed, and reliable rail track inspection. The sensitivity of detection as a function of lift-off was also scrutinised. It was concluded that detectability could be enhanced using the figure-8 shaped probe at a frequency of 0.3 MHz, while maintaining a lift-off distance of around 1 mm. The results from both case studies indicated that the Tx-Rx sensor exhibits superior SNR in detecting ferromagnetic material structures. Measurements revealed that ferromagnetic material structures are more readily detected at relatively low frequencies (0.3 MHz), facilitating in-line electronics and data acquisition. The probe with the figure-8 shaped probe demonstrated optimal SNR near the resonant frequency peak of the receiver coil. The enhanced inspection system represents an innovative and fundamentally promising application, closely aligning with the criteria requisite for integration in Industry 4.0. To fully harness the potential of this advanced inspection system and optimise its use, it is imperative to include real rail track defect characterisation and demonstrate its effectiveness in detecting manufacturing flaws in authentic environments. Consequently, future work will involve validating the effectiveness of a new EC instrument, integrated into a special inspection handcart at the Scunthorpe plant of British Steel. This future research will focus on detecting squats and other localised types of damage.

## Figures and Tables

**Figure 1 sensors-24-06702-f001:**
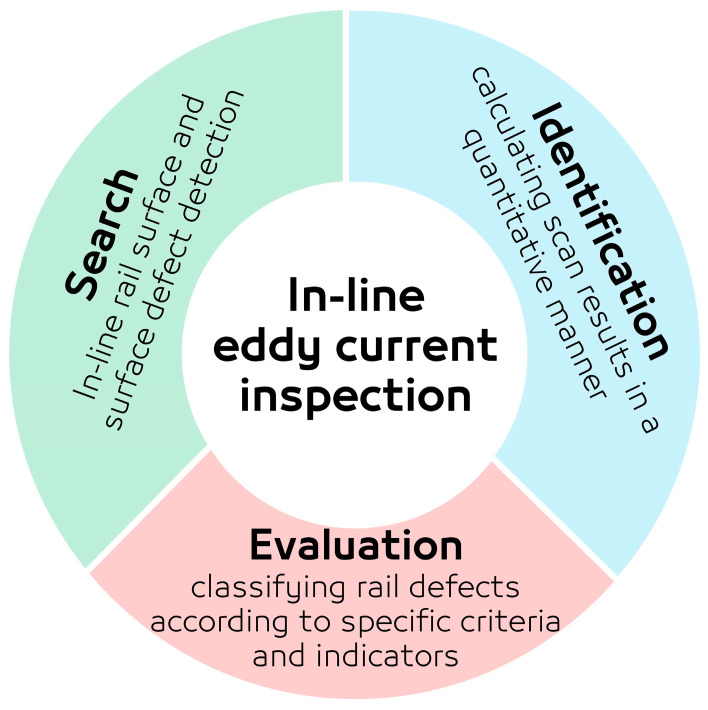
Improved efficiency of current railroad inspection.

**Figure 2 sensors-24-06702-f002:**
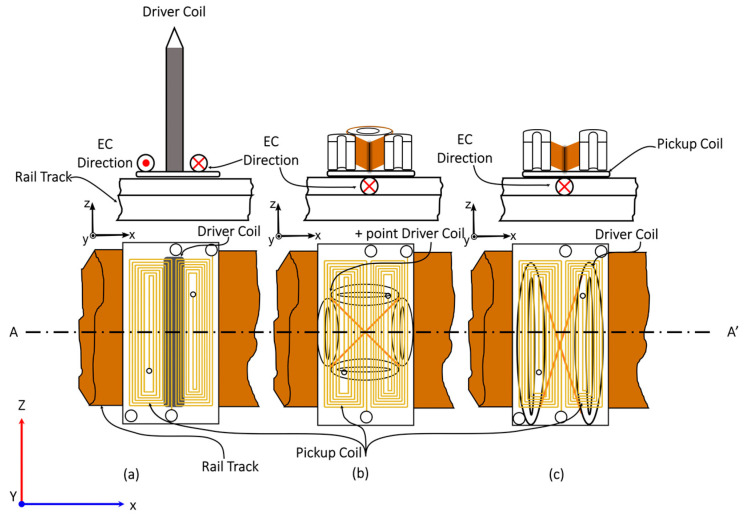
Three eddy current probe configurations with: (**a**) singular; (**b**) +point, and (**c**) figure-8 shaped.

**Figure 3 sensors-24-06702-f003:**
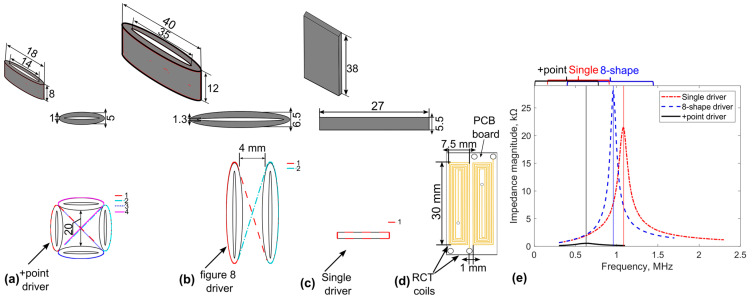
Directional EC probe design: (**a**) winding method of the +Point, (**b**) figure-8 shaped, (**c**) rectangular single transmitter, (**d**) rectangular (RCT) receiver coil dimensions (top-down view) and (**e**) resonant frequencies of different transmitter coils.

**Figure 4 sensors-24-06702-f004:**
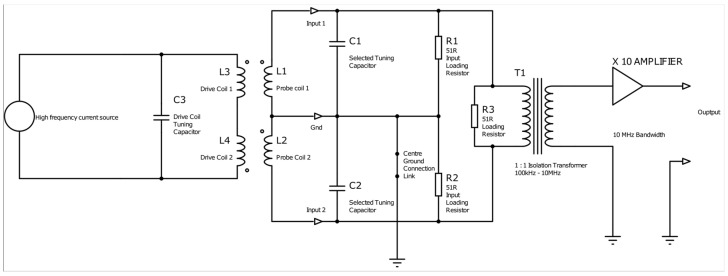
Simplified diagram of amplifier with tuning caps and drive coils.

**Figure 5 sensors-24-06702-f005:**
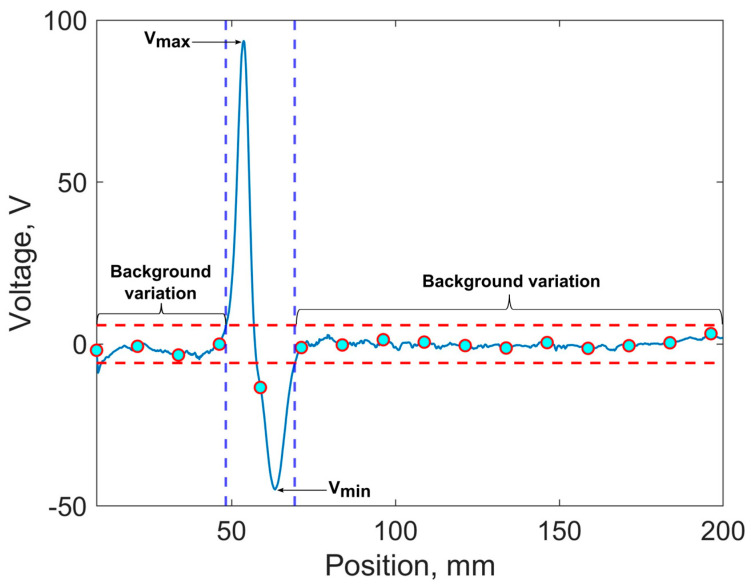
Exemplar plot of identification process of proposed system.

**Figure 6 sensors-24-06702-f006:**
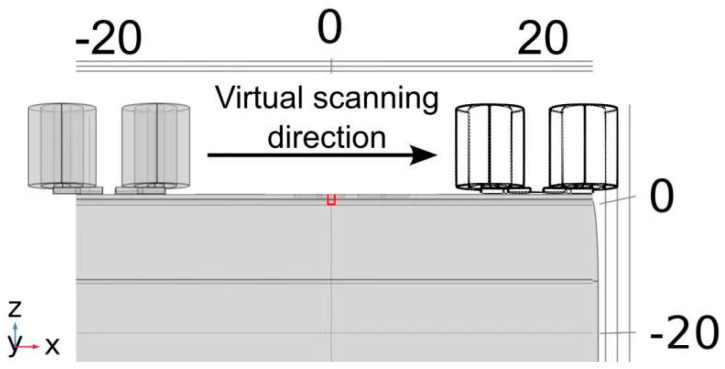
Schematic of the virtual scanning model.

**Figure 7 sensors-24-06702-f007:**
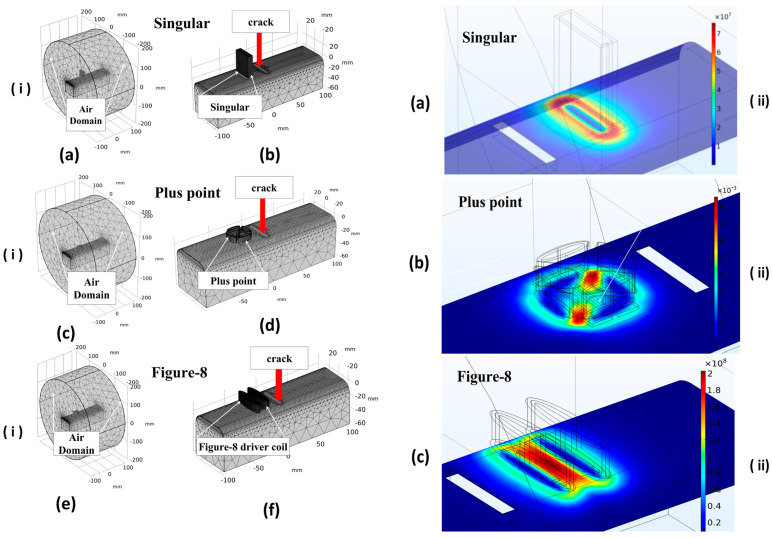
(**i**) Simulated mesh overview: (**a**,**c**,**e**) show the mesh with the air domain, while (**b**,**d**,**f**) provide a zoomed-in view of the transmitter coils’ positioning near the rail for the “Singular”, “+Point”, and “Figure-8” configurations, respectively. (**ii**) FEM simulations of induced eddy current flow patterns for the (**a**) singular, (**b**) plus-point, and (**c**) figure-8-shaped probes.

**Figure 8 sensors-24-06702-f008:**
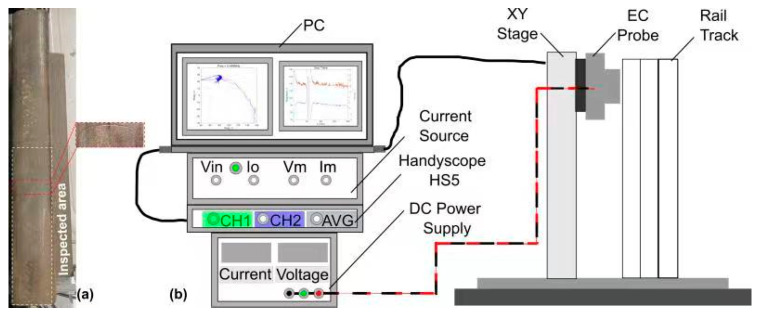
(**a**) Rail track sample (**b**). Experimental set-up.

**Figure 9 sensors-24-06702-f009:**
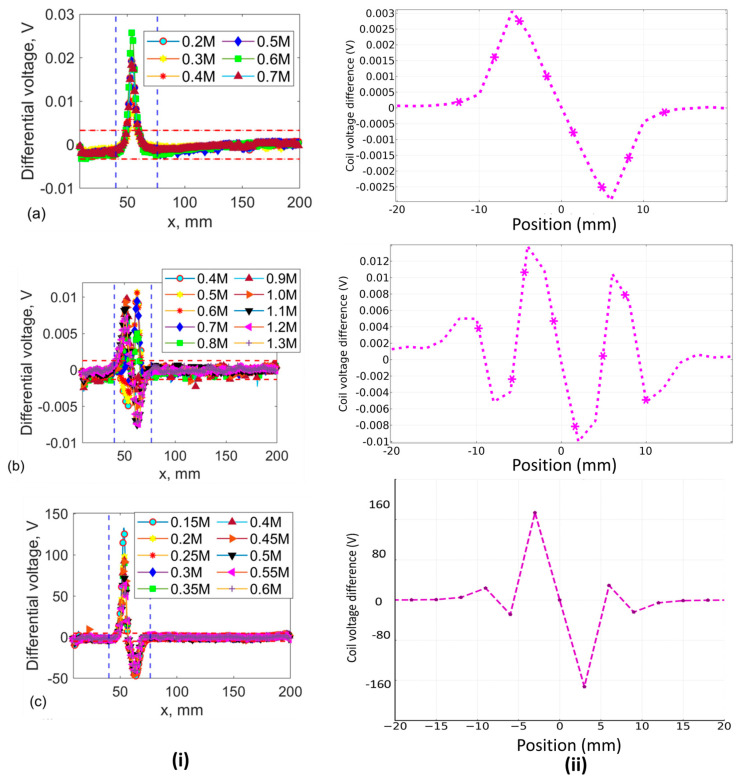
Optimal frequency selection study results depicted in rows: (**i**) experimental raw EC data; (**ii**) FEM simulations for EC probes with transmitter coils in configurations: (**a**) single, (**b**) plus-point, and (**c**) figure-8 shaped.

**Figure 10 sensors-24-06702-f010:**
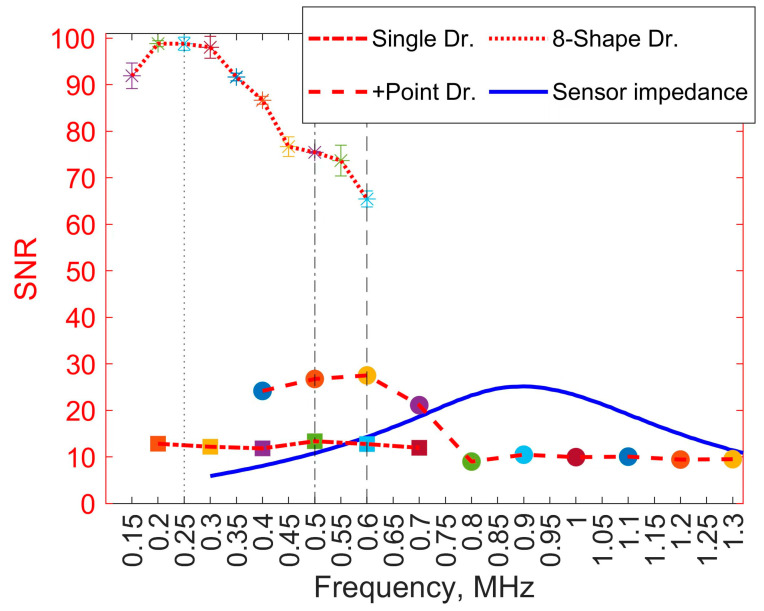
Comparative analysis of probe sensitivity across various.

**Figure 11 sensors-24-06702-f011:**
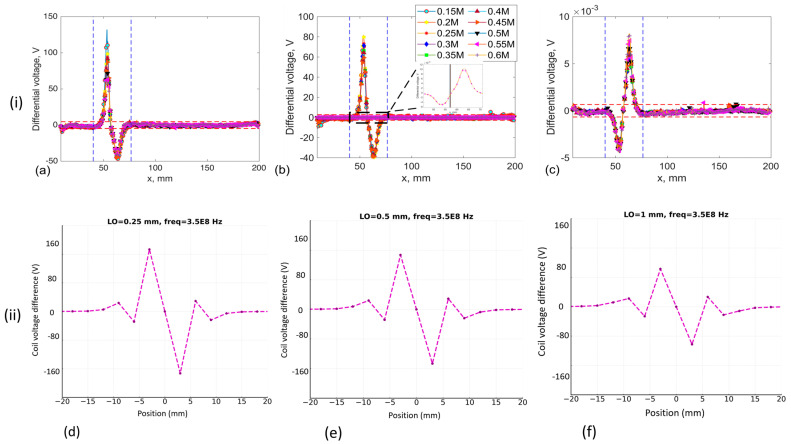
Eddy current scan data and FEM simulation results are depicted as follows: (**i**) experimental raw EC data obtained using a figure-8 shaped transmitter at lift-offs of (**a**) 0.25 mm, (**b**) 0.5 mm, and (**c**) 1 mm; (**ii**) corresponding FEM simulation outputs for the same lift-offs shown in (**d**–**f**), respectively.

**Figure 12 sensors-24-06702-f012:**
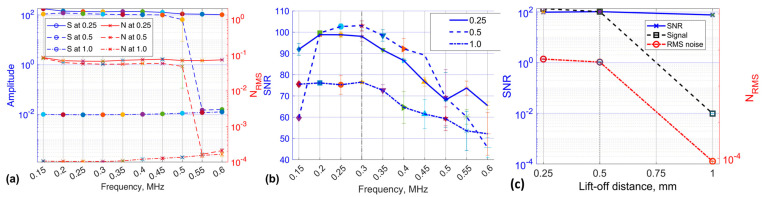
Performance assessment of EC probe with a figure-8 shaped transmitter across lift-off distances of 0.25 mm, 0.5 mm, and 1.0 mm: (**a**) separate contributions of signal (S) and noise (N) and (**b**) SNR across various frequencies; (**c**) average voltage and structural noise at optimum 0.3 MHz.

**Figure 13 sensors-24-06702-f013:**
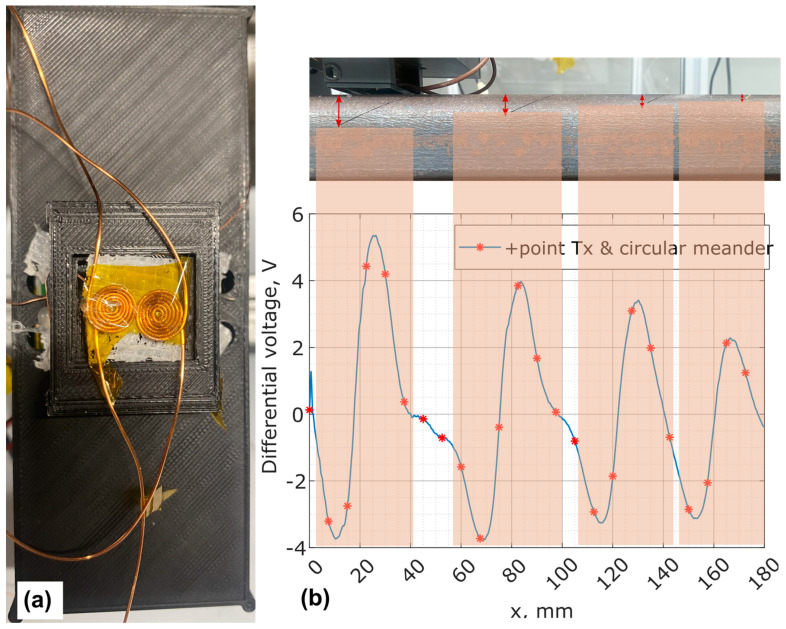
Lift-off noise insensitivity results using the initial EC probe prototype with a +Point transmitter and two differentially hand-wound meander receiver coils: (**a**) top-down view of the probe, and (**b**) side view of the probe on the sample with varying depths and EC scan results at 0.3 MHz.

**Figure 14 sensors-24-06702-f014:**
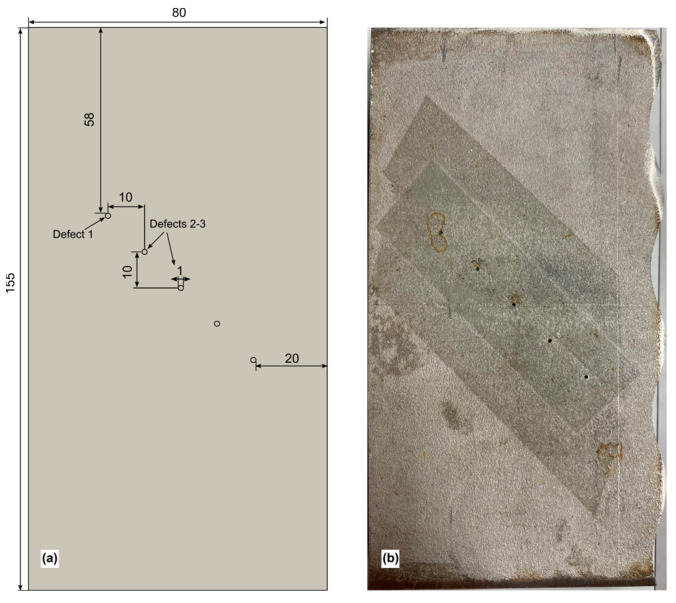
Dimensions of the specimen and artificially drilled hole: (**a**) schematic diagram; (**b**) real photo of the specimen.

**Figure 15 sensors-24-06702-f015:**
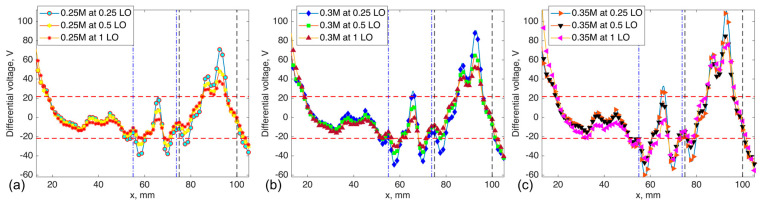
Raw EC data with lift-off distances of 0.25, 0.5, and 1 mm at (**a**) 0.25 MHz, (**b**) 0.3 MHz, and (**c**) 0.35 MHz operational frequencies.

**Figure 16 sensors-24-06702-f016:**
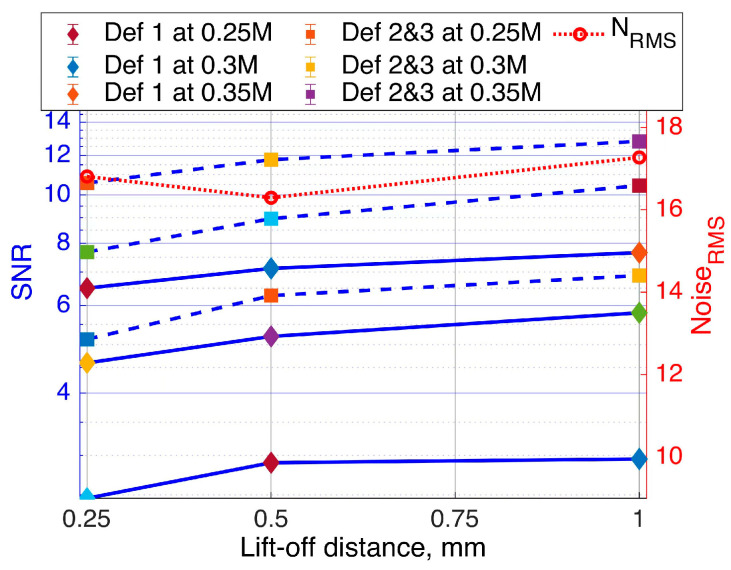
Selected probe signal-to-noise ratio over defect 1 and defects 2–3, with separate noise contributions as a function of lift-off distance.

**Figure 17 sensors-24-06702-f017:**
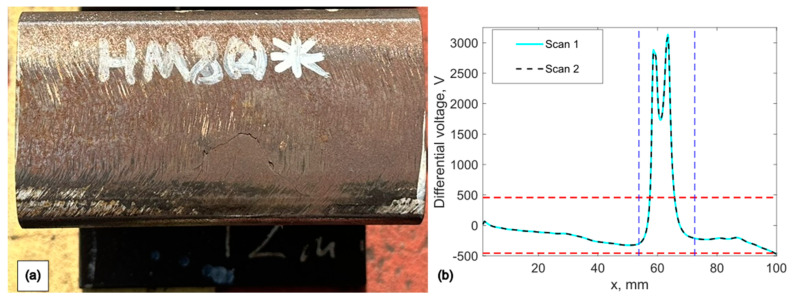
Eddy current scan results over studs using the selected probe: (**a**) inspected sample; (**b**) raw eddy current scan results from two individual scans.

**Table 1 sensors-24-06702-t001:** Material properties and dimensions of the transmitter coil.

Transmitter Coil Name	Parameters and Dimensions
Single	62 turns, 0.5 mm diameter, 38 × 27 × 5.5 mm
+point	72 turns, 0.2 mm diameter, 18 × 5 × 8 mm
Figure-8 shaped	66 turns, 0.2 mm diameter, 40 × 12 × 6.5 mm

**Table 2 sensors-24-06702-t002:** Material properties and dimensions of the sensing coil.

Receiver Coils	Parameters and Dimensions
Number of turns	12
Product family	2 layers
Copper weight	10.68 kg/m^2^
Thickness	1.6 mm
Material	FR4 (150 °C) middle Tg
Circuit size X	30 mm

**Table 3 sensors-24-06702-t003:** FEM setup parameters.

Parameters	Singular Probe	Plus-Point	Figure 8
Transmitter coil dimensions, mm	27 × 5.5 × 38	18 × 5 × 8	40 × 6.5 × 12
Transmitter wire diameter, mm	0.5	0.2	0.2
Transmitter coil turns	62	72	66
Pick up coil turns	12 turns		
Pick up coil dimensions, mm	30 × 7.5 × 1		
Pick up coil thickness	3.4 mm		
Gap between two pick-up coils	1 mm		
Pick up coil wire diameter, mm	0.2		
Lift-off distance, mm	0.25	0.25	0.25; 0.5; 1
Core material	Soft Iron		
Physics setting in COMSOL	Magnetic fields		

## Data Availability

The original data presented in the study are openly available in IEEE DataPort at https://dx.doi.org/10.21227/kraf-rf03.

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
