# Peer review of "Towards Advancing Real-Time Railroad Inspection Using a Directional Eddy Current Probe"

_sensors, 2024, doi:10.3390/s24206702_

Round 1
Reviewer 1 Report (New Reviewer)
Comments and Suggestions for Authors
While the paper has provided some useful information on rail defect detection, it does not validate the claim in the paper i.e. minimize lift-off impact. Liftoff impact is a critical issue in rail detection and the authors have provided very cursory effort on this.
In addition, the tests are not detailed , different crack depth, crack angles should be at least attempted.
In addition, there is a recent paper in the journal on rail defect detection the authors have failed to cite – which shows a comprehensive literature search is lacking.
Comments on the Quality of English Languageenglish ok
Author Response
Firstly the authors would like to thank the reviewers for their positive comments and considered suggestions. The authors have responded to each comment individually below and highlighted the changes made in the manuscript.
Reviewer 1
Comment 1: While the paper has provided some useful information on rail defect detection, it does not validate the claim in the paper i.e. minimize lift-off impact. Liftoff impact is a critical issue in rail detection and the authors have provided very cursory effort on this.
Response:
Thank you for providing the opportunity to address this query.
The SNR of the selected probe exhibits a clear downward trend as lift-off increases. The differential receiver coils used in the proposed measurement mechanism are designed not to minimize the impact of lift-off on defect sensitivity but to exploit the insensitivity of the differ-ential mode to lift-off variations, which can introduce errors in EC measurements. This means that lift-off variations can mask signals from small defects, leading to a high number of false alarms in conventional EC modes. It is important to compare SNR because high voltages can also correspond to high coherent noise. By comparing SNR results quantitatively, these com-parisons become more interpretable. As shown in Fig. 12(b), the proposed probe demonstrates better SNR performance even at a 1 mm lift-off distance.
Change: The authors have included a statement clarifying this decision in the subsection “Lift-off distance and optimum probe sensitivity” section of the manuscript.
Comment 2: In addition, the tests are not detailed , different crack depth, crack angles should be at least attempted.
Response:
Thank you for the chance to clarify this point.
4.4 Experimental validation of probe sensitivity to studs
The experimental results from railroad inspection are crucial for validating the effec-tiveness of the selected EC probe. The results presented in previous sections are based on artificially induced defects in a material with an undamaged, high-quality surface. For this study, a real rail section with local damage, measuring 128 mm in length, was pre-pared. The material has a carbon content in the range of 0.87-0.97% by mass in its liquid state. In real rail sections with induced local damage, such as studs (see Fig. 16(a)), major cracks develop in the direction of traffic, although the propagation mechanism remains under investigation. Some cracks form at approximately 20 degrees, but there is no con-sistent pattern, leading to variations in crack depth and angle along with high roughness in the undamaged areas of the specimen. The defects typically have a "U" shape and can penetrate to depths of 3-6 mm in the rail and defect formation extensively studied in [34]. Studs pose a risk because the rail beneath them cannot be effectively tested using ultra-sonic NDE from the running surface, except with specialized equipment [34].
Two repeat scans were performed on the defected sample, with the results plotted in Fig. 16(b). The proposed probe exhibits excellent sensitivity at optimum frequency to both studs and non-defected areas of the specimen, achieving SNR of 15 and 14.8, indicating high repeatability. The rough surface of the real rail sample is evident in the EC scan re-sults from 0 to 54 mm. Both receivers detected the defected zone, as indicated by the peaks between the vertical dashed lines in Fig. 16(b). The noise threshold, represented by the vertical dashed line, shows the mean RMS noise level multiplied by two.
Change: The authors have provided clarifications by creating a new subsection titled "Experimental validation of probe sensitivity to studs".
Comments 3: In addition, there is a recent paper in the journal on rail defect detection the authors have failed to cite – which shows a comprehensive literature search is lacking.
Response: Thank you for the recommendation. The following manuscripts were added to the "Introduction" and "Methodology" sections of the manuscript:
- X. Li, G. Tian, K. Li, Q. Zhang, and X. Lu, “Investigation of rolling contact fatigue cracks using the transmitter-receiver eddy current testing under moving conditions,” Nondestruct. Test. Eval., vol. 39, no. 3, pp. 614–633, 2024, doi: 10.1080/10589759.2023.2220871.
- F. Carere, A. Sardellitti, A. Bernieri, L. Ferrigno, S. Sangiovanni, and M. Laracca, “An Eddy Current Probe for the Detection of Subsuperficial Defects of Any Orientation,” IEEE Trans. Instrum. Meas., vol. 73, pp. 1–13, 2024, doi: 10.1109/TIM.2024.3395324.
Your suggestions and comments are of great value in guiding this ongoing refinement process.

Reviewer 2 Report (New Reviewer)
Comments and Suggestions for Authors
The manuscript introduces a hybrid eddy current testing probe, which can enhance detection sensitivity and minimize lift-off impact. The authors use the simulations and experiments to investigate the feasibility of the proposed probe. After reviewing the manuscript, following comments and questions has been raised to the authors:
1. Many figures in this manuscript are not clear enough, I believe they maybe unreadable. In addition, the font sizes in this manuscript are inconsistent, please modify them.
2. Which unit of copper weight in Table 2? 3. The SI units and U.S. customary units were presented in Table 3 simultaneously.
3. In section 2.8, it is recommended to describe the finite element mesh in more detail. Which finite elements were used for different regions in the FEA model? What is the number of elements in different region? Has the effect of the mesh size on the simulation results? What criterion was used for meshing? In addition, the type of defect should be specified: crack or hole?
4. Line 360, ‘In contrast, the ECs densities in Figure 7.ii.(d)’, where is Figure 7.ii.(d)?
5. In section 2.9, please specify the purpose of Probe Configuration Analysis.
6. Line 393, the contents in figures does not correspond to the titles. Similarly, Line 572, ‘(see Fig. 8(b))’, please check it.
7. Line 417: ‘0.2÷0.7 MHz’? ‘0.4÷1.3 MHz’? ‘0.15÷0.6 MHz’? Please check them.
8. Line 527: ‘Three initial defects, situated from the left edge as illustrated in Fig. 12,’, maybe in Fig. 13?
Author Response
Firstly the authors would like to thank the reviewers for their positive comments and considered suggestions. The authors have responded to each comment individually below and highlighted the changes made in the manuscript.
Reviewer 2:
Comment 1: Many figures in this manuscript are not clear enough, I believe they maybe unreadable. In addition, the font sizes in this manuscript are inconsistent, please modify them.
Change: Thank you for the recommendation. The Figures and font sizes were corrected as suggested.
Comment 2: Which unit of copper weight in Table 2? 3. The SI units and U.S. customary units were presented in Table 3 simultaneously.
.Change: It was corrected.
Comment 3: In section 2.8, it is recommended to describe the finite element mesh in more detail. Which finite elements were used for different regions in the FEA model? What is the number of elements in different region? Has the effect of the mesh size on the simulation results? What criterion was used for meshing? In addition, the type of defect should be specified: crack or hole?
Response: To ensure the accuracy of the simulation results, different finite elements were utilized for various regions within the FEM model. Fine mesh elements were used near the defect zones to capture detailed EC interactions, while coarser elements were applied to the non-defective surrounding areas to optimize computational efficiency. The number of elements varied depending on the region, with a higher concentration of elements in the critical defect areas to enhance detection precision. The mesh size had a significant impact on the simulation results. A finer mesh size increased the resolution and accuracy of the defect detection but also required more computational resources. Conversely, a coarser mesh reduced computational load but could potentially overlook smaller defect features. Therefore, a balance was achieved by adjusting the mesh density according to the specific requirements of different regions within the model.
The criterion for meshing involved ensuring that the mesh elements were sufficiently small to accurately represent the defect features. In this study, the defect type specified was a crack. This allowed for precise simulation of the EC distributions and their interactions with the crack defect. The accurate representation of cracks was essential for evaluating the sensitivity and performance of the different probe configurations. This method proved effective in increasing sensitivity and accuracy in defect detection, which is critical for practical NDT applications.
Change: The authors have provided clarifications by creating new subsections titled "2.8.1 Mesh configuration and finite elements" and “2.8.2 Defect type and meshing criteria”.
Comment 4: Line 360, ‘In contrast, the ECs densities in Figure 7.ii.(d)’, where is Figure 7.ii.(d)?.
Change: It was corrected as it was suggested.
Comment 5: In section 2.9, please specify the purpose of Probe Configuration Analysis.
Response: The primary objective of the probe configuration analysis is to rigorously assess the efficacy of various probe configurations—namely 'Singular,' '+Point,' and 'Figure-8'—in detecting defects through induced ECs. This comparative evaluation seeks to ascertain which configuration demonstrates superior sensitivity and precision in defect detection, thereby informing the optimal design for practical applications in NDT.
Change: The authors have provided clarifications in section 2.9.
Comment 6: Line 393, the contents in figures does not correspond to the titles. Similarly, Line 572, ‘(see Fig. 8(b))’, please check it.
Change: The Figures were corrected as suggested.
Comment 7: Line 417: ‘0.2÷0.7 MHz’? ‘0.4÷1.3 MHz’? ‘0.15÷0.6 MHz’? Please check them.
Change: The text was corrected as it was suggested.
Comment 8: Line 527: ‘Three initial defects, situated from the left edge as illustrated in Fig. 12,’, maybe in Fig. 13?.
Change: The text was corrected as it was suggested.
Thank you for your many helpful suggestions. We tried to address your comments.

Round 2
Reviewer 1 Report (New Reviewer)
Comments and Suggestions for Authors
The authors have not fully addressed my questions in the first round of review.
The answer on lift-off insensitivity is not convincing.
Author Response
Reviewer 1
Comment 1: While the paper has provided some useful information on rail defect detection, it does not validate the claim in the paper i.e. minimize lift-off impact. Liftoff impact is a critical issue in rail detection and the authors have provided very cursory effort on this.
Response:
Thank you for providing the opportunity to address this query.
The differential receiver coils used in the proposed measurement mechanism are designed not to minimize the impact of lift-off on defect sensitivity but to exploit the insensitivity of the differential mode to lift-off variations, which can introduce errors in EC measurements. Abdalla et al. [10] pointed out limitations of ECT related to lift-off, particularly in reflection mode: As the lift-off distance increases, the interaction between the magnetic field and the material weakens, making it more challenging for the receiver coil to detect changes in the impedance caused by the interaction of the primary magnetic field with the ECs in the material. The emf that the receiver coil detects is a result of this interaction. This issue arises because the ECs themselves become less intense at greater distances, reducing the strength of the secondary magnetic field that the receiver coil needs to detect. In addition to the experimental data, FEM simulations were conducted to assess the impact of lift-off distance on probe performance. The simulation results, presented in Fig. 11 (d), (e), and (f), demonstrate that the selected probe configuration exhibits a similar trend to the experimental results as the lift-off distance increases. A significant drop in differential voltage was observed at the maximum lift-off distance of 1 mm.
The initial EC probe design combined a +Point receiver coil with two differentially hand-wound meander pickup coils. Initial tests used circular hand-wound differential coils before finalizing the design on a PCB board with a 0.5 mm wire and a 6.5 mm coil diameter, as shown in Fig. 13(a). The measurement setup involved placing the sample on a vibrating linear stage while the EC probe remained static at 1 mm lift-off distance. Despite receiver coil design inconsistencies, preliminary results were promising. Stage vibration, caused by the sample's weight after each movement increment, allowed only one repeat measurement, as shown in Fig. 13(b). The reduction in differential voltage was attributed to variations in the depth of artificially induced defects, with the lowest voltage corresponding to a defect depth of around 1 mm. Since the same Tx-dRx measurement mechanism was applied throughout all studies, this insensitivity to lift-off variation test results demonstrates the mechanism's reliability, robustness, and effectiveness in real-world scenarios before dynamic testing.
Figure 13. Lift-off noise insensitivity results using the initial EC probe prototype with a +Point receiver and two differentially hand-wound meander pickup coils: (a) top-down view of the probe, and (b) side-view of the probe on the sample with varying depths and EC scan results at 0.3 MHz.
In addition, during the measurements, a video recording was made. Please visit this link to watch the video: https://drive.google.com/file/d/1pRsoySYtUOQT6fM-MW46xwVoydP28bDF/view?usp=sharing
Change: The authors have included a statement clarifying this decision in the subsection “Lift-off distance and optimum probe sensitivity” section of the manuscript.

Round 3
Reviewer 1 Report (New Reviewer)
Comments and Suggestions for Authors
no further comments
This manuscript is a resubmission of an earlier submission. The following is a list of the peer review reports and author responses from that submission.
Round 1
Reviewer 1 Report
Comments and Suggestions for Authors
1. The description of the structure of the 8-shape driver coil is not detailed enough;
Why can the 8-shape driver coil achieve better performance? The design concept should be introduced in principle.
3. The author needs to add an introduction to the process and signal processing of detection methods.
How real-time is this method?
5. Some images are blurry.
6. The difference in image size is too large.
Author Response
Firstly the authors would like to thank the reviewers for their positive comments and considered suggestions. The authors have responded to each comment individually below and highlighted the changes made in the manuscript.
Comment 1: The description of the structure of the 8-shape driver coil is not detailed enough.
Response:
Thank you for providing the opportunity to address this query.
The decision to utilize the 8-shaped winding driver coil over the single driver is informed by previous research by the authors in Ref. [28], indicating that the elongated figure 8-shaped coil endeavors to induce eddy currents along the preferential orientation of asymmetric rectangular coils. This phenomenon arises from the currents driven across the scanning direction, resulting in significant current densities in areas containing defects.
Change: The authors have included a statement clarifying this decision in the subsection “Driver coil design concept” section of the manuscript.
Comment 2: Why can the 8-shape driver coil achieve better performance? The design concept should be introduced in principle.
Response:
Thank you for the chance to clarify this point.
In the configuration where elongated 8-shaped coils drive current across the scanning direction, as depicted in Figure 7, we observe a clear increase in current density at the defective zone. This phenomenon arises from the generation of currents in both clockwise and counterclockwise directions. The strong magnetic field concentrates between the two elongated coils, covering the entire receiver area with a relatively uniform field, thus enhancing detectability for receivers and inducing significant disturbances in eddy current flow.
To capitalize on this concept, we have redesigned the 8-shape driver to further elongate it, thereby enhancing the induction of strong eddy currents across the scanning direction. We anticipate that any deviations from this preferred orientation will be detectable by the underlying differential sensors. As a result, we hypothesize that the elongated figure 8-shape coil may offer improved sensitivity to small local damages within the material.
Change: The authors have provided clarifications for both decisions in the subsections "Frequency selection study" and "Driver coil design concept" sections, respectively.
Comments 3: The author needs to add an introduction to the process and signal processing of detection methods.
Response: Thank you for the recommendation. The measurement mechanism proposed in this study is based on the principle of electromagnetic induction. An alternating current supplied to the coil generates a changing magnetic field around the wires, inducing eddy currents in nearby conductive materials. These induced currents create their secondary magnetic fields, flowing opposite to the primary magnetic field, and contain information about the conductivity and permeability of the tested material. By monitoring changes in receiver impedance, the presence of defects can be detected.
Your suggestions and comments are of great value in guiding this ongoing refinement process.
Change: The structure of the paper was reorganized to accommodate this, namely a subsection (“1.5. Working principle and signal processing of sensors”) is created.
Comment 4: How real-time is this method?
Response: Thank you for your question. The measurement process occurs in real-time and is monitored online via the Matlab environment, with further post-processing conducted. The real-time inspection demonstration of EC probes equipped with both point and figure 8 shape drivers can be found in the supplementary materials. This demonstration video illustrates the experimental scanning process and the detection of anomalous signals, indicating the presence of rail defects. Additional details on post-processing methods can be found in Appendix A1 of Ref. [29].
Change: The authors have included a statement clarifying this decision in the subsection “1.5. Working principle and signal processing of sensors”.
Comment 5: Some images are blurry
Response: Thank you for your observations.
Change: We have submitted all images utilized in the manuscript at high resolution (300 ppi) for the revised version, provided as a separate file.
Comment 6: The difference in image size is too large.
Response: Thank you for your many helpful suggestions.
Change: Thank you for your recommendation. We have provided all images in high quality.

Reviewer 2 Report
Comments and Suggestions for Authors
The reviewer understood that the paper focuses on the application of eddy current testing technology to the evaluation of deterioration of railroad rails. However, the impact of the novelty of the paper is weak, and the reviewer felt that the following points need to be considered and modified for acceptance.
The authors discussed lift-off and sensitivity, but how is this study positioned in relation to the current required specifications in applications field? Also, what are the advantages and improvements of the paper results over other previous papers? Although the lift-off and sensitivity are discussed for three types of sensor configurations, they are optimizations just within this paper.
The reviewer thinks it is possible to discuss the optimal frequency and sensitivity for the three configurations to some extent using the FEM simulations.
The authors used FEM simulation results as part of the verification of the experimental results, but there are discrepancies between the simulation and experimental results. The reasons suggested by the authors are inspection speed, hysteresis of the material, and coercivity, but they are not clear. How is speed reflected in the simulations? Also, how are the magnetic properties of the material affected to induced voltage at the coils? The description needs to be written in a way that can be understood.
What resonant frequency for the pick-up coil.
Why applied voltage is constant for driving coil? The impedance of the driving coil is varying depending on frequency, therefore applied field is different, which affects the sensitivity of the measurement.
Totally, more detailed explanation is required for the figure of experimental results.
Minor parts:
Page 14, line 328, nonuni90o this is ???
Not easy to recognize (or read) the characters in the figures.
Reviewer 3 Report
Comments and Suggestions for Authors
I recommend your article for publication with minimal but thorough technical correction.
My main comments relate mainly to the technical design of the text. “Sensors” magazine has a very simple and very convenient universal template for designing articles https://www.mdpi.com/files/word-templates/sensors-template.dot
Notes:
1. I would include “non-destructive testing” in the list of keywords.
2. Before line 137, insert the title of the section “2. Methodology” with subsequent changes in the numbering of sections and subsections.
3. Throughout the text, replace the separating symbol in three-dimensional dimensions “x” with the symbol “´”.
4. Throughout the text, expressions similar to “38X27X5.5 mm” should be replaced with “38´27´5.5 mm3”.
5. At the end of the names of Figure 2 and Figure 10, put “.”.
6. Lines 144 and 145 must be connected.
7. In Table 2, correct “1.6mm” to “1.6 mm”.
8. In lines 217, 218, 200, 232, separate numerical values and dimensions with a space, for example, “100 kHz”, “6 dB”.
9. In Table 3, change the line “Driver coil dimensions, mm” to “Driver coil dimensions, mm3” and “Pick up coil dimensions, mm” to “Pick up coil dimensions, mm3”.
10. A “References” section should be inserted into line 431.
11. The formatting of links should be uniform. Design of links in MDPI style. The link to https://doi.org/... makes it easier for the curious reader to access the cited article.
Wang, S.; Xia, X.; Ye, L.; Yang, B. Automatic Detection and Classification of Steel Surface Defect Using Deep Convolutional Neural Networks. Metals 2021, 11, 388. https://doi.org/10.3390/met11030388
12. Check the possibility of increasing the clarity of some figures; they can be presented in additional materials to the article in enlarged form for possible better reading of the inscriptions. The editors welcome this and it will not cause any difficulties.
Be careful when correcting the text of the article, maybe I missed something else.

Author Response
Firstly the authors would like to thank the reviewers for their positive comments and considered suggestions. The authors have responded to each comment individually below and highlighted the changes made in the manuscript.
Comment 1: I would include “non-destructive testing” in the list of keywords.
Change: Thank you for the recommendation. The keywords list was updated as suggested.
Comment 2: Before line 137, insert the title of the section “2. Methodology” with subsequent changes in the numbering of sections and subsections
.
Change: It was corrected and the subsection numbers were corrected following this section.
Comment 3: Throughout the text, replace the separating symbol in three-dimensional dimensions “x” with the symbol “.
Change: The text was formatted throughout the paper.
Comment 4: Throughout the text, expressions similar to “38X27X5.5 mm” should be replaced with “38´27´5.5 mm3”.
Change: It was replaced as it was suggested.
Comment 5: At the end of the names of Figure 2 and Figure 10, put “.”.
Change: It was corrected as it was suggested.
Comment 6: Lines 144 and 145 must be connected.
Change: The paper was formatted as suggested.
Comment 7: In Table 2, correct “1.6mm” to “1.6 mm”.
Change: The text was formatted throughout the paper.
Comment 8: In lines 217, 218, 200, 232, separate numerical values and dimensions with a space, for example, “100 kHz”, “6 dB”.
Change: The text was formatted throughout the paper as suggested.
Comment 9: In Table 3, change the line “Driver coil dimensions, mm” to “Driver coil dimensions, mm3” and “Pick up coil dimensions, mm” to “Pick up coil dimensions, mm3”
Change: The units were replaced throughout the paper as suggested.
Comment 10: A “References” section should be inserted into line 431.
Change: The “References” section was moved to correct line as specified.
Comment 11: The formatting of links should be uniform. Design of links in MDPI style. The link to https://doi.org/... makes it easier for the curious reader to access the cited article.
Wang, S.; Xia, X.; Ye, L.; Yang, B. Automatic Detection and Classification of Steel Surface Defect Using Deep Convolutional Neural Networks. Metals 2021, 11, 388. https://doi.org/10.3390/met11030388
Response: Thank you for your comments. The reference formatting links were corrected as suggested.
Change: The text was formatted throughout the paper.
Comment 12: Check the possibility of increasing the clarity of some figures; they can be presented in additional materials to the article in enlarged form for possible better reading of the inscriptions. The editors welcome this and it will not cause any difficulties.
Be careful when correcting the text of the article, maybe I missed something else.
Response: Thank you for your many helpful suggestions. We tried to address your comments.
Change: The figures were formatted throughout the paper as suggested.
